# Factors Affecting the Developmental Status of Children Aged 6 Months to 2 Years in Urban and Rural Areas

**DOI:** 10.3390/children10071214

**Published:** 2023-07-13

**Authors:** Rita Andriyani, Eddy Fadlyana, Rodman Tarigan

**Affiliations:** Child and Health Department, Faculty of Medicine, Padjadjaran University, Hasan Sadikin Hospital West Java Indonesia, Bandung 40161, Indonesia; andriyani3001@yahoo.com (R.A.); rodman.tarigan@unpad.ac.id (R.T.)

**Keywords:** children aged 6 months to 2 years, developmental delays, urban, rural

## Abstract

The first two years of life, including the period from conception to 2 years of age, are very important for a child’s growth and development. This study sought to describe the developmental status and the environmental factors that influence it in children aged 6 months to 2 years in urban and rural areas. The research sites were purposively selected: specific health centres in the city of Bandung and West Bandung district were chosen and the study was conducted from November to December 2022. A cross-sectional study was conducted with healthy children aged 6 months to 2 years and their parents, who agreed to participate in the study. Researchers performed developmental tests using the Developmental Pre-screening Questionnaire with classifications for normal developmental test results and developmental delays. During the study, 346 children met the inclusion criteria, resulting in a sample of 164 (47.4%) boys and 182 (52.6%) girls. There were no significant differences among children with developmental delays in urban and rural areas. The factors that influence the possibility of developmental delays in the two research areas were the child’s age, exposure to screen time, stimulation, nutritional status, and the use of the mother–child handbook.

## 1. Introduction

The first two years of life, known as the first thousand days of life (from 270 days of gestation until the child is two years) are simultaneously a golden period and a sensitive period for a child’s growth and development [1]. During this period, brain plasticity, or the brain’s ability to develop based on the child’s experiences, occurs. The total brain volume of a child at 1 month of age is about 36% of that of an adult brain, increasing to 72% at 1 year of age and reaching 83% at 2 years of age [2].

The factors that contribute during this period are both genetic and environmental. Good and correct parenting, including providing good and sufficient nutrition and adopting suitable feeding practices, proper stimulation, good health status, and environmental conditions, including where a child lives, grows, and develops, are very influential during this period for children to achieve optimal growth and development [1,2]. Growth is quantitative: it can be measured with standard tools and is assessed as body weight, height or length, head circumference, and upper arm circumference. Development refers to the increase in complex abilities and bodily functions that are qualitative in nature and follow a predictable pattern due to the maturation process [3]. Child development consists of several aspects, including gross motor, fine motor, language, cognitive, personal, and social skills. Developmental delays in children occur when a child fails to achieve one or more of these developmental levels as expected by a certain age, and this is influenced by socioeconomic, maternal, biological, environmental, nutritional, and genetic factors, among others [4,5]. Children’s basic needs are broadly grouped into three categories, physical–biomedical needs, needs for affection (love), and needs for training/stimulation (skill-building), which must be met early on [6].

Until now, rural and urban areas are areas of concern in various fields, including economic equality and parents’ level of education about meeting children’s needs. In addition, rural geographical conditions entail sparser population densities than urban areas, whereas urban areas have dense populations. This also differentiates the patterns of interaction in rural and urban communities [7].

Environmental factors of the areas in which the children live play an important role in influencing the growth and development of children. Stunting and poverty rates are more common in rural communities than urban areas. In addition, facilities and infrastructure in urban areas such as the provision of clean water, health services, and educational facilities are more complete than in rural areas. These illustrates that children who live in rural areas are more at risk of not getting good facilities and infrastructure to support their growth and development [7,8].

One-third of Indonesia’s population are children. There are around 80 million children in Indonesia, which has the fourth-largest child population in the world. Some children live in large cities. In urban areas, poverty and pollution are the main challenges that children face, while children in remote rural areas struggle with poverty and limited access to basic services [8].

A study by Muljati et al., conducted to analyse growth disorders and weight deficits in toddlers in urban and rural areas, showed that 42.6% of toddlers exhibited below-normal growth in urban areas and 53.8% did so in rural areas. In this study, more than half of the subjects lived in rural areas and as many as 62.9% were categorised as poor [9]. Therefore, efforts to improve nutrition should correspond with efforts to improve the economy, especially in rural communities. Other studies on the relationship between socioeconomic conditions and short stature from 24 to 60 months show that the prevalence of short stature is greater in children of families with middle to lower socioeconomic statuses. Parents’ education, weight for age, fathers’ occupations, and family incomes are significantly related to short stature [10].

This study sought to describe the developmental status and environmental factors that influence it in children aged 6 months to 2 years in urban and rural areas in the city of Bandung and the West Bandung district to identify the problems encountered and potential prevention efforts.

## 2. Materials and Methods

This research is a cross-sectional study that was conducted in the working area of the Garuda Public Health Center in Bandung City and its fostered Integrated Healthcare Center, and in the working area of the Padalarang Public Health Center in West Bandung Regency and its fostered Integrated Healthcare Center (Laksana Mekar, Cipendeuy, Kertamulya and Kertajaya), which were purposively selected.

The study was conducted from November to December 2022. The research subjects were children aged 6 months to 2 years who were healthy at the time of examination and whose parents agreed to participate in the study. The parents of subjects who came to the Public Health Center or Integrated Healthcare Center for routine weighing and immunisation were interviewed regarding the mother’s characteristics (education, occupation, family income) and the condition of the child, including how often they used screen time and provided stimulation as suggested in the mother–child handbook. Then, anthropometric measurements and developmental assessments were performed. Nutritional status was assessed using the 2006 WHO standard curve. Development was assessed using the Developmental Pre-screening Questionnaire, with results showing either that there were no developmental delays or that there were developmental delays.

The collected data were analysed with SPSS version 24.0 for Windows. The incidence of children experiencing developmental delays from each urban and rural area was calculated. Differences in the number of children with developmental delays in the two areas were tested with chi-squared tests. The factors that influence child development were analysed via logistic regression.

## 3. Results

During the study period, up to 400 children were assessed, but only 346 of them met the inclusion criteria for urban and rural areas, resulting in a sample of 164 boys and 182 girls.

### General Data on Mothers and Children in Urban and Rural Areas

The children in this study were aged 6 to 24 months and were divided into a 6- to 12-month category (45.1% of the children from urban areas and 48% of those from rural areas) and a 13- to 24-month category (54.9% of the children from urban areas and 52% of those from rural areas). The sexes of the children from cities and villages in this study were equally distributed. Up to 6.4% of children from urban areas and 8.7% of those from rural areas exhibited wasting. Stunting occurred in 26.6% of children from urban areas and 31.2% of those from rural areas. More than 80% of the mothers from urban and rural areas were housewives with less than a high school education or the equivalent, as Table 1 shows. Table 2, below, shows the relationships between variables and the results of developmental tests on children in the two study areas.

In urban areas, exposure to screen time and stimulation affect the developmental test results in a statistically significant way. In rural areas, children’s nutritional status and stimulation are significantly related to developmental delays.

Table 3 illustrates the relationships among the study variables and the results of developmental tests on the subjects in both study areas (urban and rural). In total, 21 (12.1%) of the subjects experienced developmental delays in urban areas and the remaining 152 subjects (87.9%) showed no delays. This is similar to rural areas, where 22 subjects (12.7%) out of a total of 173 subjects exhibited developmental delays; the number of subjects experiencing developmental delays in both study areas was not significantly different. This was not true of exposure to screen time, stimulation, or nutritional status, which statistically significantly influenced the incidence of developmental delays in both urban and rural areas (*p* < 0.05).

Table 4 shows the results of the logistic regression analysis of the factors associated with developmental delays in the two study areas. The factors affecting children aged 6–12 months include ≥1 h per day of exposure to screen time, stimulation that is not performed every day, malnutrition, and a lack of use of the mother–child handbook, i.e., that the mother did not bring the mother–child handbook to the visit. These factors statistically significantly influenced the incidence of developmental delays.

## 4. Discussion

This study shows that in urban areas, 21 children experienced developmental delays out of 173 children who were examined (12.1%). Rural areas showed almost the same ratio; 22 children experienced developmental delays out of a total of 173 children examined (12.7%). This is not statistically significant (*p* > 0.05) and contrasts with previous research that showed developmental delays in toddlers reaching as much as 30% in rural areas and 19% in urban areas [11]. However, the results of this study are also supported by previous research into delayed motor development in kindergarten-aged children in urban and rural areas that used the Denver II instrument and showed no significant difference between the motor development of children living in rural and urban areas [12]. Such results may have shifted the pattern of village community interaction, the similarity of economic conditions between villages and cities, and the level of parents’ education in rural areas, which began to match that of parents in urban areas. In addition, primary health care centres have been installed throughout rural areas, allowing villagers to easily access the information and health services that mothers and toddlers need. The Integrated Healthcare Center and its members in both urban and rural areas are quite active and routinely assess toddlers’ growth, as well as provide counselling on maternal and toddler health and basic immunisations administered by local midwives.

Based on the results of the normal development tests and the delays found among children from the two study areas, the bivariate analysis of the variables studied shows that in urban areas, the factors that influence development test results for children aged 6–24 months are exposure to screen time and stimulation. There was a significant difference between children who received less than 1 h per day of total screen time and those who received more than 1 h of screen time per day (*p* = 0.014). Likewise, children who received sufficient stimulation every day achieved significantly different scores than children who did not receive sufficient stimulation or did not receive stimulation every day (*p* < 0.001). These statistics affirm that the more exposure to screen time and the less time devoted to a child’s stimulation, the greater the developmental delays.

In rural areas, the statistically significant factors that affected developmental delays were nutritional status and stimulation. There was a significant difference in the developmental delays of children with normal nutrition and those who were malnourished (*p* = 0.015). The same was true of stunting. In rural areas, stunting has a significant effect on developmental delays (*p* = 0.042). As in urban areas, in rural areas stimulation affects the incidence of developmental delays (*p* < 0.001). This means that in rural areas, the lower a child’s nutritional status, the greater the presence of stunting, and the less time they are stimulated, the greater the possibility of developmental delays.

A bivariate analysis using combined data from urban and rural areas found that the variables that influenced the incidence of developmental delays regardless of location were nutritional status (*p* = 0.008), exposure to screen time (*p* = 0.007), and stimulation (*p* < 0.001), while the total number of developmental delays identified was not affected by the research area. Children living in urban and rural areas exhibited nearly the same incidence of developmental delays (*p* > 0.05). The delay rate of 12.1% in urban areas and 12.7% in rural areas has decreased compared to the results of previous studies [11]. However, these results align with other literature that states that the incidence of developmental delays in children aged 0–3 years in Indonesia is 13–18% [13]. The incidences of developmental delays in children, especially in the first thousand days of life, are crucial data as this rate may reduce the population’s future quality of life, so thinking about and attempting to overcome the factors that cause developmental delays as early as possible is crucial.

The logistic regression analysis that assessed variables in urban and rural areas showed that development is related to a child’s age, screen time, stimulation, nutritional status, and the mother’s use of the mother–child handbook. The analysis results show that these five variables strongly influence developmental delays with an accuracy of 89.9%. Children aged 6–12 months in urban and rural areas are 3.88 times more likely to experience developmental delays than children aged 13–24 months (95% CI, 1.68–8.97). This occurs because children under one year of age are still often carried wherever the mother or caregiver performs their daily activities, which naturally limits the child’s movement and affects motor development.

Children aged 6–24 months who receive more than 1 h per day of total screen time are 3.01 times more likely to experience developmental delays than children who receive a total screen time of less than 1 h per day (95% CI, 1.33–6.8). This is because children who are exposed to substantial screen time lack opportunities to interact with the outside world, play and receive age-appropriate stimulation, so their developmental abilities, especially in terms of language, will be hampered. These results agree with previous research by Perdana et al., who reported that children aged 18 months to 3 years who watched television for more than 4 h per day had a four times greater risk of experiencing language delays [14]. Another study by Fajariyah et al. showed a relationship between the intensity of device use and the development of children aged 24–60 months [15].

Stimulation also plays an important role in child development. The results of this study indicate that stimulation is significant in the occurrence of delays in children from both urban and rural areas (*p* < 0.001). Children aged 6–24 months who do not receive optimal stimulation or do not receive it every day are 24.75 times more likely to exhibit developmental delays than children who receive optimal stimulation every day. This aligns with the principle that stimulation must be conducted as often as possible, in pleasant conditions, continuously, and according to a child’s developmental age.

A study in rural China on pairs of children aged 18–30 months and their caregivers reported a large rate of developmental delays in children who were weakly attached to their caregivers and rarely received stimulation. The caregivers of children with developmental delays rarely read storybooks, sang, or played with the children [16]. Another study that used children aged 6–29 months reported an increase in cognitive abilities among children who were stimulated in the form of reading books and singing songs [17]. A study in the Bogor area of Indonesia examined the role of parental stimulation on the development of children under 5 in poor families and reported that children under 5 were not adequately stimulated for their age. Stimulation was conducted by 48–72% of families, and after 18 months of age children receive most of their stimulation from their mothers. Providing stimulation correlates with child development. None of the children in the high-stimulation category exhibited slow development [18].

Another factor that influenced children’s development in this study was nutritional status. Children with lower nutrition were 3.33 times more likely to experience developmental delays than children with normal nutrition (95% CI, 1.14–9.74). Children who experience malnutrition or chronic malnutrition and are not treated properly will experience stunting, which will further affect their cognitive development [19]. Similar results were reported by research on children aged 18–36 months in Tanzania in a study that examined the relationship between anthropometric indicators and child development. The results showed a strong association between the incidence of malnutrition and cognitive, language, and motor delays [20]. A meta-analysis of research in 29 low- to middle-income countries reported a strong association between linear growth in the first two years of life and motor and cognitive development. Effective nutrition interventions, accompanied by stimulation and knowledge about health will improve child development [21].

In this study, the use of the mother–child handbook was also assessed, although this factor was determined based on whether the mother brought the mother–child handbook to a routine check-up visit at the Integrated Healthcare Center or Public Health Center. Children whose parents or caregivers did not bring the mother–child handbook to the visit were 4.43 times more likely to exhibit developmental delays than children whose parents brought the mother–child handbook to the visit (95% CI, 1.17–16.84). Mothers or caregivers who did not bring the mother–child handbook to the visit might not feel that completing the mother–child handbook is important or might not pay attention to its contents, so they have less knowledge about child development. A cross-sectional study of rural areas conducted at the Ajibarang Health Center, Banyumas, that analysed the relationship between the function of the mother–child handbook and mothers’ knowledge of the handbook found that parents did not fully understand the use of the mother–child handbook, so they were not motivated to regard it as important [22]. A cross-sectional observational study conducted in Surabaya on 288 parents of children aged 3–72 months sought to examine the factors that correlate with the use of the mother–child handbook. The results of that study showed that the mother’s participation in an Integrated Healthcare Center was positively correlated with the use of the mother–child handbook, while the mother’s age, education level, and occupation were not related to the use of the mother–child handbook [23]. Further research is needed to investigate various aspects of mothers’ or parents’ knowledge about the use of the mother–child handbook and its relationship to the development of children under 5 so that the mother–child handbook can be used as effectively as possible.

The limitations of this study include that it used a cross-sectional design, which cannot robustly test for causal relationships. There may have been ‘recall bias’ or memory bias at the time of data collection, which required the mother or caregiver to report on their child and their actions. This study was limited to children aged 6–24 months, so the other environmental factors that influence the development of children over 2 years of age were not assessed. Mothers or caregivers were only asked about using the mother–child handbook in terms of whether they brought it to the visit. This question cannot assess in depth how parents understand and use the mother–child handbook.

## 5. Conclusions

There is no difference in the incidence of developmental delays in children aged 6–24 months in urban and rural areas. The environmental factors that affect developmental delays in children aged 6–24 months in urban and rural areas are the child’s age, screen time, stimulation, nutritional status, and the caregiver’s use of the mother–child handbook.

The findings of this study asserted the need for more integrated and targeted interventions to reduce malnutrition in Indonesia. The stunting rate has generally decreased in recent years, but this is still a joint task in efforts to reduce stunting rates according to the intended target.

The present study indicated that screen time, stimulation, and the caregiver’s use of the mother–child handbook affect the child’s developmental delays. Therefore, policies regarding screen time and the importance of stimulation need to be emphasized and redistributed to the wider community, as well as the mother–child handbook which supports monitoring of the growth and development of children in households. Parents need to receive regular counselling about the importance of knowledge of the environmental factors that affect children’s development, which may be regularly performed by local health workers.

Further research on the factors that influence the development of toddlers in urban and rural areas is needed, with a research design that can more strongly link cause and effect, such as a case–control design. Further research is also needed on other environmental factors that may influence child development in addition to those that were investigated in this study.

## Figures and Tables

**Table 1 children-10-01214-t001:** Characteristics of subjects by study area.

Area		
Characteristics	Urban *n* (%)	Rural *n* (%)
Age		
6–12 months	78 (45.1)	83 (48.0)
13–24 months	95 (54.9)	90 (52.0)
Gender		
Male	82 (47.4)	82 (47.4)
Female	91 (52.6)	91 (52.6)
Nutritional status		
Malnutrition	11 (6.4)	15 (8.7)
Normal	157 (90.8)	154 (89.0)
Overweight	5 (2.9)	4 (2.3)
Stunting		
Yes	46 (26.6)	54 (31.2)
No	127 (73.4)	119 (68.8)
Mother’s education		
Elementary	32 (18.5)	55 (31.8)
Intermediate	118 (68.2)	88 (50.9)
High	23 (13.3)	30 (17.3)
Mother’s job		
Housewife	143 (82.7)	148 (85.5)
Working mother	30 (17.3)	25 (14.5)
Family income		
<Regional minimum wage	106 (61.3)	115 (66.5)
≥Regional minimum wage	67 (38.7)	58 (33.5)

**Table 2 children-10-01214-t002:** Relationship between variables with developmental test results in the study area.

Variable	Urban			Rural		
	Delay (*n* = 21)	Normal (*n* = 152)	*p* *	Delay (*n* = 22)	Normal (*n* = 151)	*p* *
Age						
6–12 months	12 (15.4)	66 (84.6)	0.236	12 (14.5)	71 (85.5)	0.509
13–24 months	9 (9.5)	86 (90.5)		10 (11.1)	80 (88.9)	
Nutritional status						
Malnutrition	3 (27.3)	8 (72.7)	0.234	4 (26.7)	11 (73.3)	0.015 *
Normal	17 (10.8)	140 (89.2)		16 (10.4)	138 (89.6)	
Overweight	1 (20.0)	4 (80.0)		2 (50.0)	2 (50.0)	
Stunting						
Yes	3 (6.5)	43 (93.5)	0.173	11 (20.4)	43 (79.6)	0.042 *
No	18 (14.2)	109 (85.8)		11 (9.2)	108 (90.8)	
Screen time						
<1 h	9 (7.8)	106 (92.2)	0.014 *	12 (10.3)	105 (89.7)	0.160
≥1 h	12 (20.7)	46 (79.3)		10 (17.9)	46 (82.1)	
Stimulation						
Optimal	1 (0.9)	116 (99.1)	<0.001 *	4 (3.7)	105 (96.3)	<0.001 *
Not optimal	20 (35.7)	36 (64.3)		18 (28.1)	46 (71.9)	
Family income						
<Regional minimum wage	11 (10.4)	95 (89.6)	0.372	13 (11.3)	102 (88.7)	0.432
≥Regional minimum wage	10 (14.9)	57 (85.1)		9 (15.5)	49 (84.5)	
Use of mother–child handbook						
Yes	18 (11.3)	141 (88.7)	0.383	19 (11.7)	144 (88.3)	0.119
No	3 (21.4)	11 (78.6)		3 (30.0)	7 (70.0)	

Information: * based on chi-squared test results.

**Table 3 children-10-01214-t003:** Relationships between variables with developmental test results (aggregated data).

Variable	Developmental Test Results		
	Delay (*n* = 43)	Normal (*n* = 303)	*p* *
Area			
Urban	21 (12.1)	152 (87.9)	0.871
Rural	22 (12.7)	151 (87.3)	
Age			
6–12 months	24 (14.9)	137 (85.1)	0.192
13–24 months	19 (10.3)	166 (89.7)	
Nutritional status			
Malnutrition	7 (26.9)	19 (73.1)	0.008 *
Normal	33 (10.6)	278 (89.4)	
Overweight	3 (33.3)	6 (66.7)	
Stunting			
Yes	14 (14.0)	86 (86.0)	0.572
No	29 (11.8)	217 (88.2)	
Screen time			
<1 h	21 (9,1)	211 (90.9)	0.007 *
≥1 h	22 (19.3)	92 (80.7)	
Stimulation			
Optimal	5 (2.2)	221 (97.8)	<0.001 *
Not optimal	38 (31.7)	82 (68.3)	
Family income			
<Regional minimum wage	24 (10.9)	197 (89.1)	0.240
≥Regional minimum wage	19 (15.2)	106 (84.8)	
Use of mother–child handbook			
Yes	37 (11.5)	285 (88.5)	0.053
No	6 (25.0)	18 (75)	

Information: ***** based on chi-squared test results.

**Table 4 children-10-01214-t004:** Factors associated with developmental delays in children aged 6–24 months in urban and rural areas based on logistic regression.

Variable	Koef B	SE (B)	*p*	OR adj (CI 95%)
Age (6–12 months)	1.355	0.428	0.002	3.88 (1.68–8.97)
Screen time (≥1 h/day)	1.102	0.415	0.008	3.01 (1.33–6.80)
Stimulation (not optimal)	3.209	0.518	<0.001	24.75 (8.97–68.3)
Nutritional status (malnutrition)	1.203	0.548	0.028	3.33 (1.14–9.74)
Use of mother–child handbook (not carrying handbook)	1.489	0.681	0.029	4.43 (1.17–16.84)
Accuracy = 89.9% R^2^ (Nagelkerke) = 0.414				

Information: OR adj (CI 95%) = odds ratio adjusted (95% confidence interval).

## Data Availability

Data will be available on the main site of the study. Contact the author for future access.

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
