# Peer review of "Factors Affecting the Developmental Status of Children Aged 6 Months to 2 Years in Urban and Rural Areas"

_children, 2023, doi:10.3390/children10071214_

Round 1

Reviewer 1 Report

This is a very timely and needed paper as the authors investigated the developmental status and those environmental factors which may affect it in children aged 6 months to 2 years in both urban and rural areas in the city of Bandung and West Bandung district.

GENERAL COMMENTS

                     The title properly reflects the subject of the paper.

                     The abstract provides an accessible summary of the manuscript.

                     The keywords accurately reflect the content.

                     The introduction sets out the argument, summarizes recent research related to the topic, and highlights gaps in current understanding or conflicts in current knowledge.

                     The results are well discussed, and conclusions also consider the limits of the study.

                     The methods are appropriate, and the results are clearly presented.

                     The paper has an appropriate length.

                     References are balanced, updated, and complete.

                     The tables are clearly organized.

What does appear to be missing though is a description of the urban and rural areas under investigation.

I think that this information could provide a better background for the readers interested in this very nice paper.

Minor editing of English language required

Reviewer 2 Report

I congratulate the authors for an interesting work, well worked out and nicely presented. I feel the exercise is well round up and authors are also well aware of the limitations of their work (like cross-sectional study for causality).

1. Kementrian Kesehatan Republik Indonesia/Kemenkes RI. Stimulasi Deteksi dan Intervensi Dini Tumbuh Kembang Anak 292 di Tingkat Pelayanan Kesehatan Dasar. Quality. Jakarta: Kemenkes RI; 2022. 1–6 p. 293
2. Onigbanjo MT, Feigelman S. The First Year. Dalam: Marcdante KJ, Robert M. Kliegman, editors. Nelson textbook of pediatrics. 294 21st ed. Philadelphia: Elsevier; 2020. p. 1122.

As in the text the reference is with number "[1]" but at the reference list there are two references number "1" .You need to leave the one they refer to and remove the other one or, if it is an error and the second "[1]" reference is actually number 2 and number 2 is actually 3, and so on you will have to renumber them also in the text.

Reviewer 3 Report

The topic of the study is important and the paper has a huge potential in terms of policy implications. I find the paper very interesting and well-written.

However, to make the results more robust, I think deeper modelling should be applied. The reason for that is a relatively small sample (especially when data is divided into groups by important factors such as place of living).

Authors should consider models with interactions. One may expect that factors work not independently and co-occurrence of factors may matter. Therefore I recommend reporting if interactions became significant and which ones. The second issue is the possibility to apply propensity score matching (PSM) – pairing observations where the only differentiating factor is a place of living. Practice shows that PSM models avoid outliers and much better (robust) show the significance of differences – in fact, this problem is as designed for that method.

A recent stream of literature on early childhood development covers also environmental issues (e.g. https://www.mdpi.com/1660-4601/19/17/10967 ). The authors mention different environmental conditions in rural and urban areas in their paper. It should be wider discussed.

It would be very important to add a paragraph or two on the policy implications/recommendations from that study, as it should find its reflection in the activity of institutions and recommendations for mothers.

Minor issue: Line 25: Authors write “The first two years of life, known as the first thousand days of life” – it should be 3 years, as two years is ca. 700 days

Data availability: If data are to be published, it would be great to have them in open and free repositories such as Figshare etc. 

Language is OK, I do not find serious mistakes
